# Waste-Wood-Isolated Cellulose-Based Activated Carbon Paper Electrodes with Graphene Nanoplatelets for Flexible Supercapacitors

**DOI:** 10.3390/molecules28237822

**Published:** 2023-11-28

**Authors:** Jung Jae Lee, Su-Hyeong Chae, Jae Jun Lee, Min Sang Lee, Wonhyung Yoon, Lee Ku Kwac, Hong Gun Kim, Hye Kyoung Shin

**Affiliations:** 1Institute of Carbon Technology, Jeonju University, Jeonju 55069, Republic of Korea; darius1028@naver.com (J.J.L.); happyriss@naver.com (J.J.L.); cnfyddl@jj.ac.kr (M.S.L.); kwac29@jj.ac.kr (L.K.K.); hgkim@jj.ac.kr (H.G.K.); 2Department of Nano Convergence Engineering, Jeonbuk National University, Jeonju 54896, Republic of Korea; suc_0819@jbnu.ac.kr; 3Jeonbuk Institute for Regional Program Evaluation, Jeonju 54896, Republic of Korea; yoon126@irpe.or.kr

**Keywords:** supercapacitor, waste wood, cellulose fiber, graphene nanoplatelet, activated carbon paper, electrode

## Abstract

Waste wood, which has a large amount of cellulose fibers, should be transformed into useful materials for addressing environmental and resource problems. Thus, this study analyzed the application of waste wood as supercapacitor electrode material. First, cellulose fibers were extracted from waste wood and mixed with different contents of graphene nanoplatelets (GnPs) in water. Using a facile filtration method, cellulose papers with GnPs were prepared and converted into carbon papers through carbonization and then to porous activated carbon papers containing GnPs (ACP−GnP) through chemical activation processes. For the morphology of ACP−GnP, activated carbon fibers with abundant pores were formed. The increase in the amount of GnPs attached to the fiber surfaces decreased the number of pores. The Brunauer–Emmett–Teller surface areas and specific capacitance of the ACP−GnP electrodes decreased with an increase in the GnP content. However, the galvanostatic charge–discharge curves of ACPs with higher GnP contents gradually changed into triangular and linear shapes, which are associated with the capacitive performance. For example, ACP with 15 wt% GnP had a low mass transfer resistance and high charge delivery of ions, resulting in the specific capacitance value of 267 Fg^−1^ owing to micropore and mesopore formation during the activation of carbon paper.

## 1. Introduction

The continued and enormous energy consumption using fossil fuels has resulted in environmental pollution and resource depletion problems, necessitating the development of energy storage devices [1,2,3,4,5,6]. Among such devices, supercapacitors storing electrochemical energy changes, have a high power density and long lifecycle, thereby extending the operating time and life of batteries [7,8,9,10,11,12]. Supercapacitors are primarily composed of two electrodes, electrolytes, and a separation membrane that divides the two electrodes. Electrodes play an important role in supercapacitor charge storage and are categorized as electric double-layer capacitors (EDLCs) and pseudocapacitive supercapacitors based on the electrode materials [13,14,15,16,17,18]. EDLC electrodes are composed of activated or porous carbon-based materials with large surface areas that allow easy access of electrolyte ions to the electrodes for an increase in charge storage [19,20].

Recently, researchers have investigated agricultural byproducts, such as rice husks [21,22,23,24], coconut shells [25,26,27,28], or sugarcane bagasse [29,30,31,32,33], as activated carbon precursors in supercapacitor electrodes owing to their renewability, eco-friendliness, and low cost. Agricultural byproducts consist of the composition of cellulose, lignin, hemicellulose, ash, and other substances, which have different characteristic behaviors. Lignin and hemicellulose, which are amorphous structures, function as binders that cling between cellulose fibers. Meanwhile, cellulose, a semi-crystalline linear polymer, is lightweight, renewable, and also has outstanding mechanical properties owing to its high aspect ratio [34,35,36,37,38]. Considering the different characteristics of cellulose, lignin, and hemicellulose, the desired activated carbon materials should be prepared after separating these components. In addition, it is important to separate cellulose fibers as flexible porous materials because they have advantages in producing more flexible paper-shaped electrodes over hemicellulose and lignin-based porous carbon materials as a representative paper-making resource.

For example, Wang et al. [39] studied porous carbon spheres prepared from hemicelluloses discarded during the cellulose fiber extraction process for supercapacitor application. Hemicellulose-based carbon microspheres were prepared via a hydrothermal process and further activated with various activators. Among them, hemicellulose-based activated carbon spheres activated using ZnCl_2_ showed a specific capacitance of 218 F g^−1^ at 0.2 A g^−1^ in a 6 M KOH solution, but this flexible property has not been reported because they are not paper or mat. Schlee et al. [40] studied microporous carbon fiber mats for free-standing supercapacitors obtained from lignin extracted from eucalyptus. Lignin-based microporous carbon fiber mats exhibited a specific capacitance of 155 F g^−1^ at 0.1 A g^−1^ in a 6 M KOH solution, but this flexible property has not been reported. Kim et al. [24] studied flexible activated carbon paper (ACP)-shaped electrodes with high porosity prepared from cellulose fibers extracted from rice husks. Flexible ACP prepared using only cellulose fibers extracted from rice husks achieved the high specific surface area of 2158.48 m^2^ g^−1^ and specific capacitance of 255 F g^−1^ at 1 A g^−1^ in a 1 M KOH solution.

Waste wood comprises a substantial portion of cellulose fibers but has rarely been analyzed as an activated carbon material for supercapacitor electrodes because it contains impurities such as paints, adhesive, and vanishes. Nonetheless, cellulose fibers isolated from waste wood can be converted into useful materials, thereby mitigating the environmental problems caused by waste wood [41,42,43,44,45]. Therefore, activated carbon suitable for a high-capacity supercapacitor electrode from cellulose fibers extracted from waste wood needs to be investigated.

The pore size distribution and electrical conductivity of activated carbon materials are important factors for improving the electrochemical properties of EDLC [46,47,48]. In this study, cellulose fibers were extracted from waste wood in order to change waste wood into useful materials for addressing environmental and resource problems and mixed with graphene nanoplatelets (GnPs) to enhance the electrical conductivity of the resulting electrode. Subsequently, cellulose papers were prepared with different cellulose fiber and GnP contents via a facile filtration method. These papers were then converted into ACPs through carbonization and chemical activation and, consequently, used as supercapacitor electrode materials. ACPs containing various GnP contents were examined by scanning electron microscopy (SEM), Brunauer–Emmett–Teller (BET) analysis, X-ray diffraction (XRD), and Raman spectral analysis. Moreover, the potential of the results as EDLC materials was analyzed.

## 2. Results and Discussion

### 2.1. Morphology of ACP−GnP Samples

Figure 1 shows the SEM images, depicting the morphology of porous ACP containing GnPs. As shown in Figure 1a, the GnP particles are well dispersed and attached to the ACP surfaces, thereby enhancing the electrical conductivity. From the magnified SEM image in Figure 1b, the porous structures of ACP are developed with GnP particles covering and blocking the pores. An increase in the GnP content reduced the porosity of ACP. Nevertheless, the abundant pores on surfaces of the ACP surface can affect the formation of micropores and mesopores to increase the electrochemical capacitance, as shown in Figure 1c.

### 2.2. Textural Properties of ACP−GnP Samples

Porous ACP was obtained by varying the GnP content using N_2_ adsorption−desorption isotherms. As shown in Figure 2, the ACP obtained with different GnP contents comprises a combination of type I and IV isotherms with hysteresis loops according to the BET classification. As shown in Figure 2a, the N_2_ adsorption curves at a low relative pressure (*p*/*p*_0_) of <0.1 for all ACP surfaces drastically increased due to the development of micropores in ACP−GnP. Generally, when *p*/*p*_0_ approach 0.2, the N_2_ in the micropores is completely adsorbed. For *p*/*p*_0_ > 0.2, which is related to the N_2_ adsorption in the mesopores and macropores, the volume absorption of N_2_ on the ACP−GnP surfaces slowly increased owing to the existence of mesopores. However, the N_2_ adsorption curves of the ACP−GnPs decreased as the GnP content increased because the surfaces of the activated carbon cellulose fibers were covered with GnPs, as shown in the SEM images. The BET surface areas of ACP−GnP 0, ACP−GnP 1, ACP−GnP 3, ACP−GnP 5, ACP−GnP 7, and ACP−1100 were 1592.20, 1535.13, 1475.59, 1342.08, 1254. 71, and 921.78 m^2^ g^−1^, respectively.

Figure 2b shows the micropore, mesopore, and macropore distributions for all ACP−GnP samples, which were identified using the Horvath–Kawazoe and Barrett−Joyner−Halenda methods. According to pore size classification (IUPAC), the pores were divided with micropores (diameter, D < 2 nm), mesopores (2 nm ≤ D < 50 nm), and macropores (D > 50 nm). The ACP−GnP samples mostly comprise micropores and mesopores in the size range of 0–40 nm, resulting in a hierarchical pore structure. A specific surface part of ACP is increased to increase the storage capacity of the supercapacitor electrodes. Therefore, pore structures with micropores and mesopores significantly influence the electrical performance. In particular, micropores increase the high surface area, thereby increasing the storage capacity through the formation of electric double layers. Moreover, these allow easy ion transfer pathways. Therefore, the development of microporous and mesoporous structures in ACP−GnP samples plays an important role in enhancing their electrochemical capacity as energy storage materials.

### 2.3. Crystallinity of ACP−GnP Samples

The XRD patterns of the ACP−GnP samples are shown in Figure 3a, where two representative diffraction peaks appeared at 2θ of approximately 24°–26° corresponding to the (002) of all the ACP−GnP samples, respectively. These peaks are related to the crystalline structures of the carbon materials. In particular, the broad peaks at 2θ = 24–26° are associated with ACP, and the strong and sharp peaks at 26° are related to GnP. All ACPs showed similar full-width at half maximum values and intensities because all waste wood cellulose papers were treated at the same activation and carbonization temperature. Meanwhile, the peak intensities at 2θ of 26° for the GnPs with high crystallinity sharply increased with the increasing GnP content [49,50]. In the Raman spectra in Figure 3b, this characteristic is observed in the two characteristic peaks at 1351 cm^−1^ of the D band, which is related to the disordered or defective graphite structure, and 1610 cm^−1^ of the G band, which is associated with the ordered layered graphite structure. As shown in Figure 3b, the G band intensities increased with the increasing GnP content. The ratio for the intensities of the D and G band (*I*_G_/*I*_D_) were applied to determine the degree of graphitization [51,52]. The *I*_G_/*I*_D_ values are 0.97, 1.04, 1.10, 1.15, 1.18, 1.33, and 1.59 for ACP−GnP 0, ACP−GnP 1, ACP−GnP 3, ACP−GnP 5, ACP−GnP 7, and ACP−GnP 15, respectively. An increase in the GnP content increased the degree of crystallinity in ACP, thereby improving the good electrical conductivity and electrochemical performance between the ACP−GnP and electrolyte [53,54].

### 2.4. Electrochemical Performance of ACPs

The electrochemical performances of the ACPs with various GnP contents as electrodes were evaluated in a three-electrode system using 1 M KOH as the electrolyte. The measurements were conducted within the potential range of −1 V to 0 V at a potential scan rate of 100 mVs^−1^. The cyclic voltammetry (CV) curves of the ACPs with different GnP contents are shown in Figure 4a. All the curves of the ACPs with different GnP contents showed rice seed-like shapes [55]. As the GnP content increased, the current peaks of the ACPs were increased (in the order of ACP−GnP 15 > ACP−GnP 7 > ACP−GnP 5 > ACP−GnP 3 > ACP−GnP 1 > ACP−GnP 0) owing to the good electrical conductivity of the GnPs. However, the specific capacitance of the ACPs decreased with the increase in the GnP content, as shown in Figure 4b, presenting the galvanostatic charge–discharge (GCD) measurements at a current density of 1 A g^−1^ for the ACP electrodes with different GnP contents. Along with the energy storage performance, the specific capacitance resulted in the GnPs attaching to the fibers and blocking the pores of ACPs. However, the GCD curves of the ACPs with high GnP contents changed into triangular and linear forms with respect to the capacitive performance. The symmetrical and triangular GCD curves demonstrate advisable capacitive conduction and great reversibility, indicating low mass transfer resistance and high charge delivery of ions in the porous ACP−GnP samples. The specific capacitances of the ACP electrodes with various GnP content were obtained using the formula:(1)$$Cs=\frac{I△t}{m△V},$$
where *I* is the discharge current, △*t* is the discharge time, *m* is the mass of the ACP-GnP electrode, and △*V* is the potential window [24]. At 1 A g^−1^ in 1 M KOH solution, the specific capacitance values of the ACP−GnP 0, ACP−GnP 1, ACP−GnP 3, ACP−GnP 5, ACP−GnP 7, and ACP−GnP 15 electrodes were approximately 425, 411, 398, 3857, 370, and 267 Fg^−1^.

Figure 4c presented the Nyquist impedance spectrum of the ACP−GnP samples in the frequency range of 0.01–100 kHz. The plots are composed of semicircles in the high-frequency range and slopes in the low-frequency range. The semicircles are related to the resistance of the electrolyte (Rs) and charge-transfer resistance (R*_ct_*) between the ACP−GnP electrode surfaces and electrolyte. The slopes are attributed to the ion-diffusion resistance in the electrolyte. As shown in Figure 4c,d, as the electrical conductivities increased with an increase in the GnP content, the Rs values of ACP−GnP 0, ACP−GnP 1, ACP−GnP 3, ACP−GnP 5, ACP−GnP 7, and ACP−GnP 15 electrodes decreased as 2.41, 2.24, 1.98, 1.45, 1.21, and 0.62 Ω, respectively. The semicircles shrank, and the slopes increased against the *X*-axis, resulting in the faster charge transfer of the electrolyte ions into the porous ACP-GnP electrode. Although the increase in the GnP content decreased the specific capacitance of the electrode samples, the GnP content increased the number of the good ion-diffusion pathways and displayed symmetrical and triangular GCD curves. However, the ACP electrode sample with the highest GnP content (15 wt%) displayed a high specific capacitance of 267 Fg^−1^ owing to the development of micropores and mesopores during the activation of carbon paper.

## 3. Materials and Methods

### 3.1. Materials

Plywood was sourced from a waste wood treatment plant situated at Jeonju University (Jeonju-si, Republic of Korea). GnP was sourced from Nanografi Nano Technology (Çankaya, Turkey). All chemicals were of analytical grade and used as received.

### 3.2. Preparation of the Activated Carbon Paper Containing Varying GnP Content

Waste wood chips (5 × 5 cm^2^) were alkali-cooked in 15 wt% sodium hydroxide solution for 5 h at 121 °C using an autoclave. After washing using distilled water, alkali-cooked waste woods were first bleached in 2 wt% sodium chlorite and 3 wt% acetic acid solution at 70 °C for 90 min and then bleached a second time in 1.2 wt% sodium hypochlorite solution at room temperature for 60 min. The bleached pulps obtained from a two-step bleaching process were used as cellulose fibers. Waste wood cellulose fibers and GnPs were mixed in water according to the weight percentage. Subsequently, 1 wt% polyacrylamide solution was added as a binder, followed by sonication for 1 h to disperse the GnPs. The papers were prepared by a facile filtration method of the waste wood cellulose fibers and GnP solution with specific weight percentages. The obtained papers were carbonized at 900 °C under a pure N_2_ (99.999%) atmosphere. The carbon papers were immersed in NiCl_2_ solution (15 wt%) for 1 h and dried at 80 °C to prepare porous ACPs. High-thermal treatment for activation was carried out at 1000 °C under N_2_ atmosphere. After the activation, all papers were washed with H_2_SO_4_ (0.1 M) to eliminate surplus NiCl_2_ and neutralized with distilled water. Subsequently, the samples were dried at 80 °C. The ACPs containing the various GnP contents were labeled ACP−GnP 0, ACP−GnP 1, ACP−GnP 3, ACP−GnP 5, ACP−GnP 7, and ACP−GnP 15 according to their GnP content in wt%. Figure 5 displays the schematic of the preparation of ACP-GnP samples from waste wood.

### 3.3. Characterization

The morphologies of the ACP-GnP samples were observed by SEM (Hitachi SU-70, Tokyo, Japan). The specific area was evaluated using BET analysis, and the total pore volume was determined from the N_2_ adsorption data at *p/p_0_* of 0.99. Nonlocal density functional theory (NLDFT) was used to estimate the pore size distributions. The crystallinity of the ACP−-GnP samples was estimated by XRD (Rigaku, D/MAX-2500 instrument, Tokyo, Japan), with CuKα radiation at 40 kV and 30 mA, and Raman spectra (Aramis, Horiba Jobin Yvon, Tokyo, Japan) analysis.

### 3.4. Electrochemical Measurements

CV and GCD measurements of the electrochemical performance of the ACP−GnP electrodes were carried out using a CHI 660E electrochemical workstation (CH Instruments, Inc., Beijing, China). A three-electrode system with a 1 M KOH solution as the electrolyte, ACP-GnP as the electrode, an Ag/AgCl electrode as the reference electrode, and a Pt electrode as the counter electrode was used. The CV curves were obtained in the potential range of −1 to 0 V with the potential scan rates at −100 mV s^−1^ and step size 0.5 mV. GCD tests were conducted with the step-wise increase in the current density by at 1 A g^−1^ in the voltage range of −1 to 0 V vs. Ag/AgCl. Electrochemical impedance spectroscopy was performed in the frequency range of 100 kHz to 0.01 Hz to study the Rs, R*_ct_*, and ion-diffusion resistance in the electrolyte.

## 4. Conclusions

This study demonstrated the conversion of cellulose fibers extracted from waste wood into useful materials. The obtained cellulose fibers were mixed with different GnP contents in water, and the cellulose papers with different GnPs content were obtained by a facile filtration method. Subsequently, porous ACP−GnP samples for application as supercapacitor electrode materials were realized by carbonization and chemical activation. The activated carbon fibers exhibited the formation of abundant pores. An increase in the GnP content attached to the fiber surfaces decreased the number of pores. The BET surface areas and specific capacitance of ACP−GnP electrodes decreased with an increase in the GnP content. However, the GCD curves of ACPs with higher GnP contents gradually changed into triangular and linear shapes, which were associated with their capacitive performance. For example, the GCD curves of ACP−GnP 15 were approximately symmetrical and triangular line, indicating its low mass transfer resistance and high charge delivery. As such, a high specific capacitance of 267 Fg^−1^ was achieved owing to the micropores and mesopores formed during the activation of carbon paper.

Therefore, this work highlighted the use of waste wood as an active carbon source for electrode materials, thereby contributing to addressing resource and environmental problems.

## Figures and Tables

**Figure 1 molecules-28-07822-f001:**
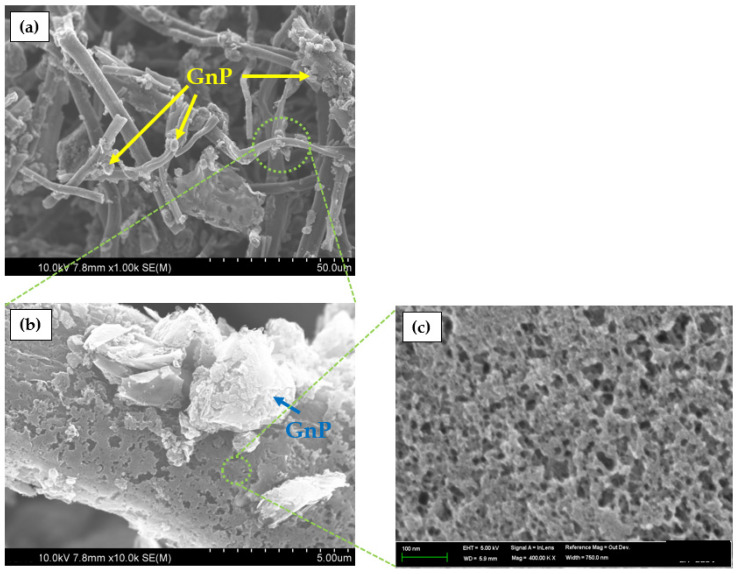
SEM images of ACP−GnP 15 at different magnifications: (**a**) 1000×, (**b**) 10,000×, (**c**) 200,000× ACP−GnP 15.

**Figure 2 molecules-28-07822-f002:**
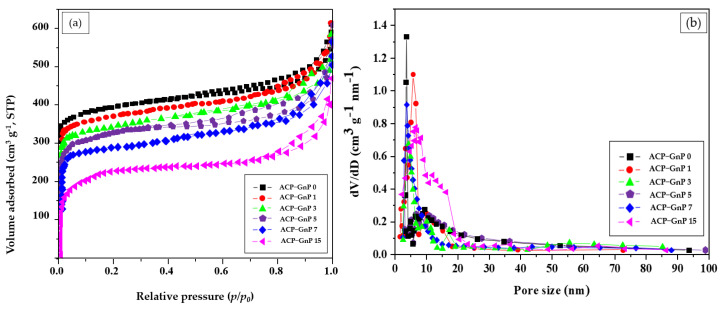
ACP N_2_ adsorption and microporosity. (**a**) Experimentally measured adsorption−desorption isotherms and (**b**) computed micropore size distributions.

**Figure 3 molecules-28-07822-f003:**
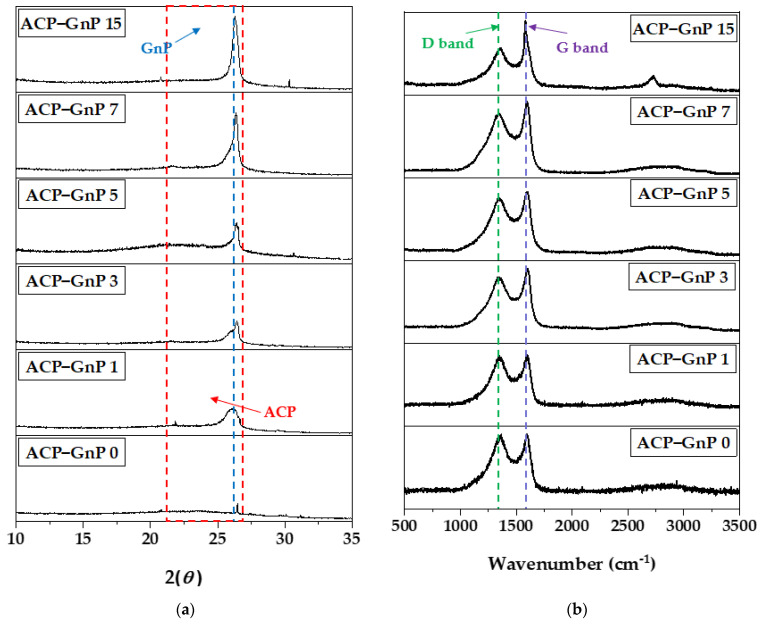
Crystal structure characterization of the ACP−GnP samples. (**a**) XRD patterns and (**b**) Raman spectra.

**Figure 4 molecules-28-07822-f004:**
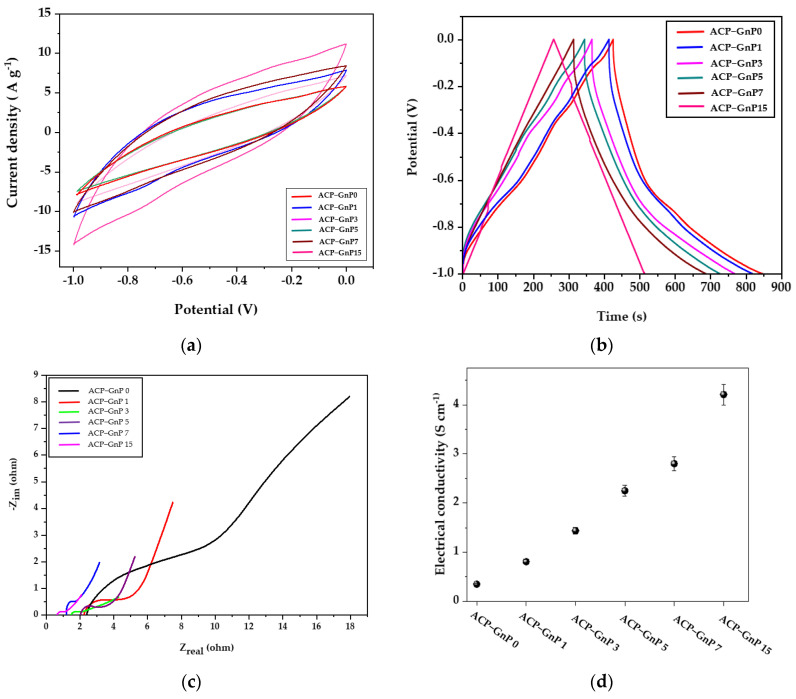
(**a**) Cyclic voltammograms at a scan rate of 100 mVs^−1^, (**b**) GCD profiles at a current density of 1 A g^−1^, (**c**) Nyquist plots t in 1 mol L^−1^ KOH, and (**d**) the electrical conductivities of the ACP electrodes containing various GnP contents.

**Figure 5 molecules-28-07822-f005:**
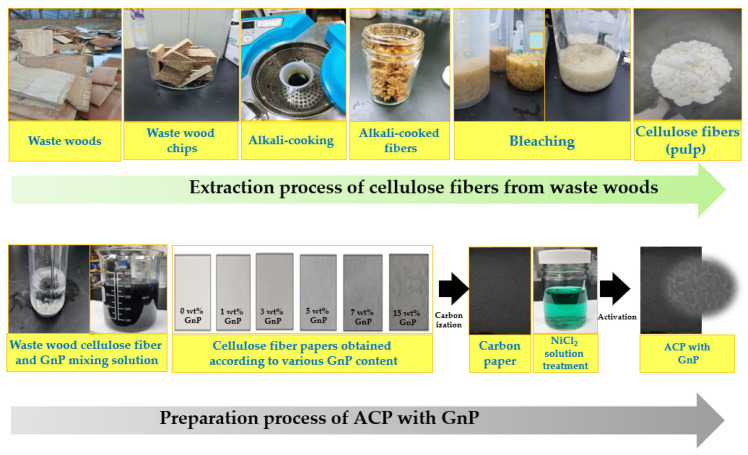
Schematic and photographs of the preparation of ACP with GnP.

## Data Availability

Data are contained within the article.

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
