# Peer review of "Waste-Wood-Isolated Cellulose-Based Activated Carbon Paper Electrodes with Graphene Nanoplatelets for Flexible Supercapacitors"

_molecules, 2023, doi:10.3390/molecules28237822_

Round 1

Reviewer 1 Report

Comments and Suggestions for Authors

You analyze the application of waste wood as supercapacitor electrode material. Cellulose fibers were mixed with different contents of graphene nanoplatelets (GnPs) in water. The cellulose papers with GnPs were converted into carbon porous activated carbon papers containing GnPs (ACP−GnP) through chemical activation processes. The increase in the amount of GnPs decreased the number of pores. The Brunauer-Emmett-Teller surface areas and specific capacitance of the ACP−GnP electrodes decreased with an increase in the GnP content. However, the galvanostatic charge-discharge curves of ACPs with higher GnP contents gradually changed into triangular and linear shapes, which are associated with the capacitive performance.

The methods are described in a good way, "at the beginning some structure investigsation such as  e-microscopy, the porosity and x-ray measurement. Then the author has a look to electrical properties." The effects are discussed and the explanations for the results are founded because it was shown that such materials can be used for electrical applications" Threrefore the conclusion is clear. The reference are up todate. Overall it is a good paper. "Therefore I asses this paper for minor corections because I have only three remarks:

-          Please add the supplier of all chemicals especially your GnP

-          Please check the number of your subheadings: 2.1 exist twice; 2.2 and 2.3 is in the chapter 3

-     The carbonization temperature is only 900 °C not 9000 °C in 3.2

Author Response

- Please add the supplier of all chemicals especially your GnP

Response: We have added a sentence to clarify the sources as follows: Plywoods were sourced from a waste wood treatment plant situated at Jeonju University (Jeonju-si, Korea). GnP was sourced from Nanografi Nano Technology (Çankaya, Turkey).

-  Please check the number of your subheadings: 2.1 exist twice; 2.2 and 2.3 is in the chapter 3

Response: The numbering of subheadings has been corrected in the revised manuscript.

- The carbonization temperature is only 900 °C not 9000 °C in 3.2

Response: We have corrected it to 900 °C

Reviewer 2 Report

Comments and Suggestions for Authors

The manuscript "Waste-wood isolated cellulose-based activated carbon paper electrodes with graphene nanoplatelets for flexible supercapacitors" delves into a highly pertinent subject in materials science and energy storage. By utilizing waste-wood isolated cellulose-based activated carbon paper electrodes with graphene nanoplatelets, this research addresses both the demand for eco-friendly materials and the need for versatile, high-performance energy storage devices.

While the manuscript demonstrates strong analytical methods that underpin the study's results, there are areas in which further clarification is required to enhance the reader's understanding. Additionally, the introduction section of the manuscript needs of significant improvement. A robust introduction is crucial to provide a clear context and background for the research, helping readers grasp the significance of the work and its place in the broader scientific landscape. Refining this section will likely contribute to the manuscript's overall quality and reader engagement.

Please find the notes below. 

Line 6. Should after "and" be one more author?  

Line 34. Are the supercapacitors among the renewable energy sources? It sounds a bit weird.

Line 46. Consist of chemical composition? Please, correct this sentence.

Line 52. Typo: hemicellulosE

Line 53. Cellulose?

Line 54-56. Please, paraphrase the sentence to make it clearer.

Lines 59-63. Paraphrase the sentence to make it clearer. Large amount of cellulose fibers seems not the best collocation. Big share of cellulose is probably better. The part about its use WITH paint, adhesive... is unclear. Do you mean that cellulose extracted from wood waste is used AS paint, adhesive....? 

Further, in the next sentence, what is meant by "various cellulose fibers"? They (fibers?) play an important role in converting of what into a useful material?

Lines 63-65. Please, try paraphrasing the sentence to make it more logically connected with the others. For now, it seems that (very simplified) cellulose from waste wood is good and various, it is used in many fields, but not in supercapacitors. Therefore, it should be investigated. There are a number of other areas where cellulose fibers are not used. Shouldn't they be investigated too?

Line 66. Pore growth? Please, check it.

Lines 67-75. Probably it should be better to claim the aim of the study, rather than to repeat the abstract one more time?

Figure 2. I would recommend choosing a better scale for the graph b. For now,  everything is mixed in the area 0-40 nm.

Line 122. What is the origin of 2 theta 43 degree?

Lines 122-129. I do not see peaks at 24-26 degrees, maybe choosing a better scale will be helpful? I see the peaks strictly at 26 degrees. Are there any references from literature to confirm that GnP gives the peak at 26 degrees?

Lines 130-131, please provide the references to confirm the interpretation of the Raman spectroscopy results.

Line 134. Please, provide the reference for the calculation of the degree of graphitization.

Line 157. "which was (Figure 1)"? Not clear.

Provide the reference for formula (1). Where are the results of the calculations?

Figure 4, b. What is marked with black dotted lines?

Line 193. Shouldn't it be plywood?

Provide doi for all the references.

The first reference has a doubled number.

Author Response

Line 6. Should after "and" be one more author?  

Response: ‘and’ has been inserted before the name of the final author.

Line 34. Are the supercapacitors among the renewable energy sources? It sounds a bit weird.

Response: My apologies for the poor phrasing that has given this impression. The relevant sentence has been revised as follows: The continued and enormous energy consumption using fossil fuels has resulted in environmental pollution and resource depletion problems, necessitating the development of energy storage devices

Line 46. Consist of chemical composition? Please, correct this sentence.

Response: The relevant sentence has been corrected to fix this error.

Line 52. Typo: hemicellulose

Response: We have corrected this error.

Line 53. Cellulose?

Response: We have replaced “cellulose” by ‘cellulose fibers’.

Line 54-56. Please, paraphrase the sentence to make it clearer.

Response: We have added a sentence as follows: In addition, it is important to separate cellulose fibers as flexible porous materials because they have advantages for producing more flexible paper-shaped electrodes over hemicellulose and lignin-based porous carbon materials as representative paper-making resource.

Lines 59-63. Paraphrase the sentence to make it clearer. Large amount of cellulose fibers seems not the best collocation. Big share of cellulose is probably better.

Response: Based on your comment, we revised “large amount of cellulose fibers” to “substantial portion of cellulose fibers”

The part about its use WITH paint, adhesive... is unclear. Do you mean that cellulose extracted from wood waste is used AS paint, adhesive....? 

Response: We have revised the relevant sentence as follows “owing to its use with paint, adhesive, wax, and other materials.” is revised to “because it contains impurities such as paints, adhesive, and vanishes.”

Further, in the next sentence, what is meant by "various cellulose fibers"? They (fibers?) play an important role in converting of what into a useful material?

Response: We have revised the relevant sentence as: Nonetheless, cellulose fibers isolated from waste wood can be converted into useful materials, thereby mitigating the environmental problems caused by waste wood.

Lines 63-65. Please, try paraphrasing the sentence to make it more logically connected with the others. For now, it seems that (very simplified) cellulose from waste wood is good and various, it is used in many fields, but not in supercapacitors. Therefore, it should be investigated. There are a number of other areas where cellulose fibers are not used. Shouldn't they be investigated too?

Response: We have added the following sentence in lines: For example, Wang et al [39] studied porous carbon spheres prepared from hemicelluloses discarded during the cellulose fiber extraction process for supercapacitor application. Hemicellulose-based carbon microspheres were prepared via a hydrothermal process and further activated with various activators. Among them, hemicellulose-based activated carbon spheres activated using ZnCl2 showed a specific capacitance of 218 F g1 at 0.2 A g-1 in a 6 M KOH solution, but this flexible property has not been reported because they are not paper or mat. Schlee et al [40] studied microporous carbon fiber mats for free-standing supercapacitors obtained from lignin extracted from eucalyptus. Lignin-based microporous carbon fiber mats exhibited a specific capacitance of 155 F g1 at 0.1 A g-1 in a 6 M KOH solution, but this flexible property has not been reported. Kim et al [24] studied flexible activated carbon paper (ACP)-shaped electrodes with high porosity prepared from cellulose fibers extracted from rice husks. Flexible ACP prepared using only cellulose fibers extracted from rice husks achieved the high specific surface area of 2158.48 m2 g1 and specific capacitance of 255 F g1 at 1 A g-1 in a 1 M KOH solution.

Line 66. Pore growth? Please, check it.

Response: We have revised “pore growth” into “pore size distribution” to address this.

Lines 67-75. Probably it should be better to claim the aim of the study, rather than to repeat the abstract one more time?

Response: We have revised the sentence as follows: In this study, cellulose fibers were extracted from waste wood in order to change waste wood into useful materials for addressing environmental and resource problems and mixed with graphene nanoplatelets (GnPs) to enhance the electrical conductivity of the resulting electrode.

Figure 2. I would recommend choosing a better scale for the graph b. For now,  everything is mixed in the area 0-40 nm.

Response: We have revised “0.5−50 nm” to “0−40 nm”

Line 122. What is the origin of 2 theta 43 degree?

Response: We have deleted 2 theta 43 degree in the revised version of the manuscript.

Lines 122-129. I do not see peaks at 24-26 degrees, maybe choosing a better scale will be helpful? I see the peaks strictly at 26 degrees. Are there any references from literature to confirm that GnP gives the peak at 26 degrees?

Response: We have added references 49 and 50 in this regard.

Lines 130-131, please provide the references to confirm the interpretation of the Raman spectroscopy results.

Response: We have added references 53 and 54 in this regard.

Line 134. Please, provide the reference for the calculation of the degree of graphitization.

Response: We have added references 51 and 52 in this regard.

Line 157. "which was (Figure 1)"? Not clear.

Response: We have deleted the words “which was (Figure 1)”.

Provide the reference for formula (1).

Response: We have added references 24 in this regard.

Where are the results of the calculations?

Response: We have added the results of the calculations in the revised manuscript, as follows: At 1 A g-1 in 1 M KOH solution, the specific capacitance values of the ACP−GnP 0, ACP−GnP 1, ACP−GnP 3, ACP−GnP 5, ACP−GnP 7, and ACP−GnP 15 electrodes were approximately 425, 411, 398, 3857, 370, and 267 Fg1.

Figure 4, b. What is marked with black dotted lines?

Response: We have deleted the black dotted lines from Figure 4(b).

Line 193. Shouldn't it be plywood?

Response: Yes, we revised it into “Plywoods”

Provide doi for all the references.

Response: Based on your comment, we added DOI for all the references.

The first reference has a doubled number.

Response: This has been fixed in the revised manuscript.

Round 2

Reviewer 2 Report

Comments and Suggestions for Authors

The authors have revised the manuscript according to the reviewer's comments. It can be published now.